# Teacher support for generative AI use and design learning outcomes in fashion design education: A structural equation modeling study

Zhiyi Zhang ⓘ *

School of Design, University of Leeds, Leeds, United Kingdom

* zoeyzhag@gmail.com

## Abstract

The rapid integration of generative artificial intelligence (AI) in creative education necessitates empirical investigation of support mechanisms facilitating effective technology adoption and learning outcomes. This study examines the structural relationships among teacher support dimensions, generative AI usage behavior, creative design capability, and design learning outcomes in fashion design education contexts. Drawing on Technology Acceptance Model and Social Cognitive Theory, we developed and tested a chain mediation model using structural equation modeling with data from 500 fashion design students across 18 Chinese universities. Results revealed differential associations among teacher support dimensions and AI usage behavior, with technical support and pedagogical support demonstrating significant predictive relationships, while emotional support showed no significant direct association. AI usage behavior was significantly associated with creative design capability, which in turn predicted design learning outcomes. Mediation analysis revealed significant indirect pathways, with 62.3% of the total association between technical support and learning outcomes mediated through behavioral and cognitive variables. The findings challenge monolithic conceptualizations of educational support, demonstrating patterns wherein instrumental competencies exhibit stronger associations with AI tool adoption than affective dimensions. The validated chain mediation structure elucidates relationship pathways through which environmental support covaries with learning outcomes via behavioral activation and cognitive development. These findings inform evidence-based strategies for AI integration in creative education, emphasizing technical infrastructure development, sustained engagement frameworks, and process-oriented assessment approaches. The study contributes to technology-enhanced learning theory by establishing boundary conditions for support effectiveness in emerging technological contexts while providing actionable guidance for educational practitioners navigating the digital transformation of design education.

**Data availability statement:** The minimal dataset underlying the findings of this study is provided as Supporting information. All data have been fully anonymised, and any institutional or contextual identifiers have been removed to protect participant confidentiality. The dataset contains only the variables required to reproduce the analyses reported in this manuscript.

**Funding:** The author(s) received no specific funding for this work.

**Competing interests:** The authors have declared that no competing interests exist.

# 1. Introduction

## 1.1 Research background

The rapid proliferation of generative artificial intelligence (GenAI) technologies has fundamentally transformed higher education landscapes, presenting both unprecedented opportunities and complex challenges for pedagogical innovation. Walczak and Cellary [1] argue that widespread access to generative AI tools necessitates reconceptualizing educational paradigms, particularly in disciplines bridging creative expression and technological application. Fashion design education has emerged as a critical domain for investigating this technological disruption, as AI-powered tools increasingly redefine the boundaries of creative practice [2].

Contemporary GenAI systems encompass multiple architectural paradigms. These include generative adversarial networks (GANs), GPT architectures, autoencoders, diffusion models, and transformers, has created a new technological ecosystem that demands careful pedagogical consideration [3]. Each architecture offers distinct capabilities for generating novel design concepts, facilitating rapid prototyping, and augmenting creative workflows. Empirical evidence indicates that GenAI tools can enhance both technical competencies and creative outcomes in educational settings when integrated through structured pedagogical frameworks emphasizing human-AI collaboration [4].

Fashion design education, as a discipline inherently situated at the intersection of artistic creativity and technical precision, faces unique challenges in navigating this digital transformation. Jung and Suh [5] demonstrate that generative AI integration in sustainable fashion textile design education enhances technical proficiencies alongside critical soft skills essential for contemporary practice. However, Purnama, Tajuddin, and Shariff [6] identify significant gaps between traditional curriculum structures and digitalized industry demands. Salolainen, Leppisaari, and Niinimaki [7] document longitudinal patterns of resistance, adaptation, and innovation characterizing how textile thinking and design expression have evolved in response to digital tools.

The technological ecosystem surrounding GenAI has expanded substantially. García-Peñalvo and Vázquez-Ingelmo [8] provide systematic mapping of GenAI evolution, identifying key trends defining current implementations. Kalota [9] emphasizes that understanding these varied architectures is essential for effective pedagogical integration. The versatility of these technologies extends beyond traditional creative disciplines, with Seth et al. [10] demonstrate successful GPT-4 implementation in surgical education, while Xu and Jiang [11] explore multimedia AI applications in art design pedagogy. This cross-disciplinary diffusion suggests that insights from diverse fields can inform fashion design education.

Digital tools in educational settings perform four fundamental functions that transcend disciplinary boundaries: information dissemination, interaction facilitation, process optimization, and outcome evaluation [12]. Within this framework, Maiden et al. [13] examine how digital tools can augment human creative thinking, providing empirical evidence from elite sports coaching that translates to design education contexts. Their subsequent work [14] further elaborates on designing creativity-enhancing

digital tools, emphasizing the importance of human-centered approaches in technology integration. These cross-disciplinary perspectives collectively indicate that successful GenAI integration requires thoughtful consideration of how tools fundamentally reshape creative processes and learning paradigms, extending beyond mere technical implementation.

Teacher support constitutes a critical mechanism facilitating technology adoption in educational contexts. Educational technology literature extensively documents this role, yet specific examination of generative AI implementation in design education remains limited. Contemporary research conceptualizes teacher support as a multidimensional construct encompassing technical, pedagogical, and emotional dimensions. Each dimension contributes uniquely to student technology acceptance and utilization patterns. The technology Acceptance Model (TAM) and its extensions provide theoretical grounding for understanding how external support mechanisms influence perceived usefulness and ease of use, ultimately shaping behavioral intentions and actual usage patterns in educational contexts.

## 1.2 Research problem

Despite the growing adoption of generative AI tools in fashion design education, significant gaps persist in our understanding of how teacher support mechanisms influence student engagement with these technologies and subsequent learning outcomes. While previous research has established the importance of technological infrastructure and curriculum design, the mediating role of teacher support in facilitating effective AI tool utilization remains empirically underexamined. This research gap is particularly problematic given evidence suggesting that inadequate support structures may lead to superficial tool adoption, limiting the transformative potential of AI technologies in fostering creative design capabilities.

The relationship between generative AI usage behavior and creative design capability development presents another critical area requiring systematic investigation. Although theoretical frameworks suggest that AI tools can augment human creativity through expanded ideation spaces and rapid iteration capabilities, empirical evidence regarding the mechanisms through which these tools translate into enhanced creative competencies remains limited. Furthermore, the pathway from creative capability enhancement to measurable learning outcomes in fashion design education has not been adequately modeled, leaving educators without evidence-based guidance for curriculum development and assessment design.

Current literature reveals three interconnected research questions that warrant systematic investigation: First, through what mechanisms does teacher support influence students' generative AI usage behaviors in fashion design education contexts? Second, how does engagement with generative AI tools contribute to the development of creative design capabilities? Third, what mediating processes connect creative design capability development to observable design learning outcomes?

This study aims to empirically investigate the structural relationships among teacher support, generative AI usage behavior, creative design capability, and design learning outcomes in fashion design education through a comprehensive structural equation modeling approach. By developing and testing a theoretically grounded model that captures these complex interrelationships, this research seeks to provide actionable insights for educational practitioners, policymakers, and technology developers working at the intersection of AI and design education.

The theoretical significance of this research lies in its contribution to extending technology acceptance theory within creative educational domains. By integrating perspectives from social cognitive theory with established technology adoption frameworks, this study advances our understanding of how environmental support mechanisms interact with individual cognitive processes to shape technology-mediated learning outcomes. The proposed chain mediation model, linking teacher support to learning outcomes through AI usage behavior and creative capability development, offers a novel theoretical framework for understanding technology integration in creative disciplines. Zhang, Zhao, and El Haddad [15] emphasize the critical need for such theoretical developments, noting that existing models inadequately capture the unique dynamics of AI-enhanced creative education.

From a practical perspective, this research provides evidence-based guidance for educational institutions seeking to optimize their generative AI integration strategies. The identification of specific support mechanisms that facilitate effective



AI tool adoption can inform professional development programs, curriculum design initiatives, and resource allocation decisions. Wu et al. [16] highlight the importance of understanding key factors that shape self-directed learning in AI-enhanced educational environments, suggesting that insights from this study could significantly enhance institutional capacity for supporting student success in technology-rich learning contexts.

The policy implications of this research extend beyond individual institutions to inform broader educational technology governance frameworks. As educational systems globally grapple with the challenges and opportunities presented by generative AI, empirical evidence regarding effective support structures and their impact on learning outcomes becomes increasingly valuable for policy formulation. This study's findings can contribute to the development of evidence-based guidelines for AI integration in creative education, ensuring that technological adoption enhances rather than disrupts pedagogical objectives while maintaining ethical standards and promoting equitable access to educational opportunities.

## 2. Theoretical Framework and Hypotheses

### 2.1 Theoretical Foundation

This study synthesizes Technology Acceptance Model (TAM) and Social Cognitive Theory (SCT) to construct a comprehensive theoretical framework for understanding generative AI integration in fashion design education. TAM, originally proposed by Davis [17], posits that technology adoption behaviors are primarily determined by two cognitive beliefs: perceived usefulness and perceived ease of use. These perceptions are influenced by external variables, ultimately shaping behavioral intentions and actual system usage. Shannaq [18] extends TAM to contemporary educational contexts, demonstrating that instructor satisfaction with technology integration significantly correlates with the quality of support provided to students, suggesting a cascading effect of technology acceptance from educators to learners. However, TAM does not specify how such environmental factors operate on users' cognitive evaluations, limiting its explanatory depth in complex educational contexts.

Recent extensions of TAM highlight the relevance of instructional environments in shaping technology acceptance. Shannaq [18] demonstrates that instructors' satisfaction with technology integration is positively associated with the quality of support provided to students, suggesting a cascading process whereby educators' acceptance attitudes translate into learners' adoption conditions. Nonetheless, within TAM, these external influences remain theoretically underspecified, particularly with regard to the psychological mechanisms linking environmental support to cognitive appraisals.

Social Cognitive Theory addresses this limitation by conceptualising learning as a process of reciprocal interaction among environmental factors, personal cognitions, and behavioural engagement [19]. Within educational technology contexts, teacher support constitutes a central environmental influence that shapes students' self-efficacy beliefs—defined as individuals' confidence in their capability to successfully perform specific tasks—and related behavioural patterns. SCT posits that environmental inputs do not directly determine behaviour; rather, they exert influence through cognitive mediators such as efficacy beliefs and outcome expectations. Integrating SCT with TAM therefore enables a more precise explanation of how instructional environments shape perceived usefulness and perceived ease of use through identifiable psychological pathways.

Within this integrated framework, teacher technical support functions as an environmental intervention that reduces perceived task complexity and enhances perceived ease of use. When instructors demonstrate operational proficiency, provide troubleshooting guidance, and model effective AI usage strategies, students gain mastery experiences that strengthen efficacy beliefs, leading to more favourable evaluations of tool accessibility. This mechanism reflects SCT's emphasis on guided mastery and observational learning. Empirical evidence from online entrepreneurship education supports this pathway, showing that instructor support predicts both perceived usefulness and self-efficacy through scaffolded learning experiences [20].

Teacher pedagogical support operates through a complementary pathway by strengthening perceived usefulness. By explicitly linking generative AI tools to learning objectives, assessment criteria, and design workflows, instructors shape

students' outcome expectations regarding the instrumental value of AI engagement. This dimension of support exemplifies SCT's proposition that environmental structures influence behaviour indirectly by shaping cognitive appraisals rather than through direct instruction. Pedagogical integration thus bridges teacher actions with TAM's core cognitive construct of perceived usefulness.

Self-efficacy occupies a central mediating position within this integrated TAM–SCT framework. While traditional TAM models largely attribute perceived ease of use to system characteristics, SCT emphasises that efficacy beliefs substantially condition how individuals interpret task demands. Students with higher AI self-efficacy are more likely to perceive generative tools as manageable and accessible, even when interacting with identical technical interfaces. Recent research synthesising TAM with SCT and related motivational theories demonstrates superior explanatory power for predicting sustained technology engagement, particularly in complex digital learning environments [21]. This enhanced predictive validity arises from SCT's capacity to specify how environmental support translates into motivational and cognitive states measured within TAM.

Emotional support, the third dimension of teacher support examined in this study, addresses affective barriers to technology adoption that traditional TAM formulations inadequately capture. Within the SCT framework, emotional support functions as an environmental resource that mitigates anxiety, validates experimentation, and fosters psychological safety—conditions that facilitate the mastery experiences through which self-efficacy develops. Ashkanani et al. [22] provide empirical evidence that these support dimensions operate synergistically, with deficiencies in any dimension potentially undermining overall technology adoption success. The emotional support dimension thus complements technical and pedagogical support by addressing affective prerequisites for the efficacy-building processes through which environmental resources translate into behavioral engagement.

Broader educational research further suggests that teacher support influences learning outcomes through multiple psychological mechanisms rather than singular direct pathways. Studies across technology-enhanced learning contexts demonstrate that perceived teacher support affects engagement and achievement through mediators such as academic buoyancy, burnout reduction, empathy development, and sustained motivation [23–25]. Although these findings originate primarily from language learning contexts, they offer important theoretical insight into how environmental support functions by simultaneously strengthening competence beliefs and buffering against affective strain—mechanisms that are directly relevant to AI-mediated creative education.

Empirical evidence from digital learning environments further reinforces this multidimensional perspective. Mozammel et al. [26] demonstrate that teacher support significantly enhances student engagement through the mediating mechanism of self-directed learning, suggesting that support structures must facilitate learner autonomy rather than dependency. This finding aligns with An, Yu, and Xi [27], who establish that perceived teacher support influences learning engagement through dual pathways of technology acceptance and learning motivation, highlighting the multiplicative effects of well-designed support systems. The COVID-19 pandemic provided an unprecedented context for examining these relationships, with Utvaer et al. [28] documenting how teacher and peer support became critical determinants of nursing students' emotional state and perceived competence during remote learning, insights that extend to design education contexts where hands-on practice is essential.

The sustainability of technology-enhanced learning depends significantly on continued engagement, as He and Li [29] demonstrate through their integrated TAM-SDT model examining mobile learning continuance intentions for second language acquisition. Their findings reveal that autonomy support, competence support, and relatedness support from teachers collectively influence sustained technology use through intrinsic motivation pathways. In fashion design education specifically, Hameed and Mimirinis [30] provide evidence from digital textile and studio-based courses underscores the necessity of carefully structured teacher facilitation to bridge traditional craft knowledge with emerging digital competencie. Collectively, these studies suggest that teacher support in AI-enhanced creative education must extend beyond technical instruction to encompass pedagogical alignment, emotional scaffolding, and metacognitive guidance, while preserving creative autonomy.



Drawing on these theoretical and empirical insights, the integrated TAM–SCT framework operationalised in this study conceptualises teacher technical, pedagogical, and emotional support as environmental factors that shape students' self-efficacy beliefs and cognitive appraisals—perceived usefulness and perceived ease of use—which subsequently relate to generative AI usage behaviour. By specifying the psychological mechanisms through which instructional support is associated with technology engagement, this framework advances beyond generic acceptance models and provides a theoretically parsimonious yet conceptually rich foundation for analysing AI integration in fashion design education.

## 2.2  Conceptual model and variable relationships

The proposed conceptual model articulates a chain mediation structure: Teacher Support → Generative AI Usage Behavior → Creative Design Capability → Design Learning Outcomes. This sequential pathway reflects the temporal and causal logic of technology-enhanced learning processes in creative disciplines. Teacher support operates as the exogenous variable, initiating a cascade of effects through behavioral and cognitive mediators toward ultimate learning outcomes.

Teacher support directly influences generative AI usage behavior through multiple mechanisms. Technical support reduces perceived complexity and enhances perceived ease of use, lowering barriers to initial adoption. Pedagogical support enhances perceived usefulness by demonstrating relevant applications and connecting tool capabilities to learning objectives. Emotional support addresses technology anxiety and builds confidence, facilitating sustained engagement. The multidimensional nature of support ensures comprehensive scaffolding throughout the adoption process, from initial exposure through advanced utilization.

Generative AI usage behavior serves as the primary behavioral mediator, transforming environmental support into active engagement with technological tools. This construct encompasses frequency of use, diversity of applications, depth of engagement, and integration into design workflows. Contemporary research on mobile learning continuance intentions reveals that technology usage behavior mediates between external support and capability development, with sustained engagement necessary for skill acquisition and competency development [21]. The specificity of generative AI tools— requiring understanding of prompting techniques, iteration strategies, and output evaluation—necessitates sustained, deliberate practice facilitated by comprehensive support structures.

Creative design capability represents the cognitive outcome of sustained AI tool engagement, encompassing enhanced ideation fluency, expanded solution spaces, and improved iteration efficiency. This construct bridges behavioral engagement with learning outcomes, capturing the transformation of tool usage into transferable competencies. The relationship between AI usage and creative capability is theorized as transformative rather than additive—AI tools fundamentally alter design processes rather than merely accelerating existing workflows. Research on Web 2.0 technologies in science education provides analogous evidence, demonstrating that interactive digital tools enhance higher-order thinking skills through mechanisms distinct from traditional instructional approaches [31].

Design learning outcomes constitute the terminal endogenous variable, representing observable manifestations of enhanced capabilities. These outcomes encompass portfolio quality, project complexity, technical proficiency, and professional readiness. The pathway from creative capability to learning outcomes reflects the application of enhanced competencies to concrete design challenges. Evaluating the intention to use ICT collaborative tools reveals that learning outcomes depend not merely on tool usage but on the cognitive transformations enabled by sustained, supported engagement [32].

## 2.3  Research hypotheses

Based on the theoretical framework and empirical evidence, this study proposes five primary hypotheses examining direct and mediated relationships among constructs.

H1: Teacher support positively influences students' generative AI usage behavior. Comprehensive teacher support reduces adoption barriers and enhances engagement motivation. The Interplay of School Readiness and Teacher Readiness demonstrates that instructor preparedness and support provision significantly predict student technology utilization patterns [33]. In generative AI contexts, where tools present novel interaction paradigms, teacher support becomes particularly crucial for overcoming initial adoption resistance.

H2: Generative AI usage behavior positively influences creative design capability. Sustained engagement with AI tools expands creative possibilities and enhances design competencies. Regular interaction with generative systems develops prompt engineering skills, aesthetic judgment for AI outputs, and hybrid human-AI workflow optimization. This hypothesis aligns with findings from structural equation modeling studies demonstrating positive relationships between technology usage intensity and capability development in educational contexts [34].

H3: Creative design capability positively influences design learning outcomes. Enhanced creative capabilities translate into superior project outcomes and portfolio development. Students with developed AI-augmented design skills demonstrate increased innovation, efficiency, and complexity in their work. This direct relationship reflects the application of enhanced competencies to evaluative contexts.

H4: Generative AI usage behavior mediates the relationship between teacher support and creative design capability. Teacher support indirectly enhances creative capabilities through its effect on usage behavior. This mediation pathway suggests that support must translate into actual tool engagement to impact capability development. The behavioral activation mechanism ensures that environmental resources manifest as individual competencies through active participation.

H5: Creative design capability mediates the relationship between AI usage behavior and design learning outcomes. The impact of AI tool usage on learning outcomes operates through capability enhancement rather than direct effects. This mediation emphasizes that mere tool usage without capability development yields limited learning benefits. The cognitive transformation mechanism ensures that behavioral engagement produces meaningful educational outcomes through internalized competencies.

These hypotheses collectively articulate a process model wherein environmental support initiates behavioral engagement, which develops cognitive capabilities, ultimately manifesting as enhanced learning outcomes. This chain mediation structure captures the complexity of technology-enhanced learning while maintaining theoretical parsimony and empirical testability.

## 3. Methodology

### 3.1 Research design

This study employs a cross-sectional survey design with structural equation modeling (SEM) to examine the hypothesized relationships among teacher support, generative AI usage behavior, creative design capability, and design learning outcomes. The selection of SEM methodology aligns with contemporary approaches to modeling complex multivariate relationships in educational research, enabling simultaneous examination of measurement and structural models while accounting for measurement error [35]. The cross-sectional design captures phenomena at a specific temporal point, appropriate for initial theory testing and model validation in emerging technological contexts.

The epistemological foundation adopts a post-positivist paradigm, acknowledging the probabilistic nature of social phenomena while maintaining commitment to empirical rigor. This stance recognizes that systematic observation and statistical analysis can approximate underlying population parameters with quantifiable precision. The methodological approach

integrates variable-centered analysis through SEM with recognition of contextual factors that may influence observed relationships.

SEM offers distinct advantages for testing the proposed chain mediation model. Unlike traditional regression approaches, SEM simultaneously estimates multiple dependent relationships while explicitly modeling measurement error. The ability to specify latent constructs measured by multiple indicators enhances construct validity and reliability. Furthermore, SEM's capacity for testing complex mediation pathways through bootstrapping procedures provides robust evaluation of indirect effects central to the theoretical model [36]. The approach enables examination of both direct and indirect pathways within a unified analytical framework, essential for understanding the cascading effects hypothesized in the conceptual model.

### 3.2  Sample and data collection

The target population comprises undergraduate and graduate students enrolled in fashion design programs at comprehensive universities in China. The sampling frame includes institutions offering accredited fashion design degrees with documented integration of digital design technologies in their curricula. Sample size determination followed established SEM guidelines recommending a minimum ratio of observations to estimated parameters. Power analysis indicated that the target sample size would provide adequate statistical power for detecting medium effect sizes while accounting for potential missing data and multivariate non-normality [37].

A stratified random sampling strategy ensures representation across academic levels and institutional types. Stratification variables include year of study (freshman through graduate), institution tier (Project 985, Project 211, and provincial universities), and geographic region (eastern, central, western China). This approach addresses potential heterogeneity in technology resources and pedagogical approaches across different institutional contexts. The stratification matrix ensures proportional allocation based on national enrollment statistics for fashion design programs, maximizing external validity while maintaining feasibility.

Data collection was conducted using an online survey platform during the spring semester of 2024. This approach enabled broad geographic coverage while ensuring standardized administration procedures. Survey distribution occurred through institutional coordinators who forwarded invitations to eligible students via official academic communication channels. Multiple reminder notifications were deployed at systematic intervals to optimize response rates. Incentive structures included entry into drawings for digital design resources, calibrated to encourage participation without introducing undue influence.

This study was conducted in accordance with ethical research standards and received formal approval from the Arts, Humanities and Cultures Research Ethics Committee at the University of Leeds (Approval Reference: FAHC 22–010, approved November 1, 2022). The ethics approval covered the broader doctoral research program examining fashion design education in China, of which this generative AI study constitutes a specific component within the approved research timeline spanning 2022–2025. All research procedures were guided by the approved ethical protocols, and Institutional Review Board approval was secured prior to the commencement of data collection.

The survey introduction explicitly outlined voluntary participation, confidentiality safeguards, and data usage limitations. Participants provided informed consent electronically by selecting an acknowledgement option before accessing the survey content. No minors were included in the study, and therefore no parental or guardian consent was required. Data anonymization procedures were implemented immediately upon collection, with identifying information stored separately from response data and protected through secure encryption protocols. Participants were assured that their individual responses would remain confidential and would only be reported in aggregate form. These procedures align with international standards for ethical research conduct while addressing specific considerations for research involving educational technologies [38].

### 3.3 Measurement instruments

The survey instrument operationalizes six latent constructs through multi-item scales adapted from established instruments, utilizing five-point Likert-type response formats anchored from 1 (strongly disagree) to 5 (strongly agree). Scale adaptation followed systematic procedures including translation-back translation protocols with expert panel review to ensure conceptual equivalence and cultural appropriateness for the Chinese educational context [39].

Teacher support measurement employs a three-dimensional conceptualization encompassing technical, pedagogical, and emotional support dimensions, operationalized through 15 items (5 items per dimension). Technical support items assess instructors' proficiency with generative AI tools and capacity to provide technical guidance (e.g., "My instructor demonstrates strong operational skills with AI design tools", "My instructor effectively troubleshoots technical problems when I encounter difficulties with AI platforms"). Pedagogical support items evaluate integration of AI tools into instructional design, curriculum alignment, and scaffolding of learning activities (e.g., "My instructor clearly explains how AI tools connect to our learning objectives", "My instructor provides structured guidance on integrating AI outputs into design projects"). Emotional support items measure encouragement provision, anxiety mitigation, and creation of psychologically safe environments for technological experimentation (e.g., "My instructor encourages me to experiment with AI tools without fear of making mistakes", "My instructor validates my efforts in learning to use AI design tools"). This multidimensional operationalization captures the comprehensive support mechanisms identified in the theoretical framework, with items adapted from established teacher support scales in technology-enhanced learning contexts.

Generative AI usage behavior measurement adapts established technology usage scales to specifically address generative AI applications in design contexts through 5 items. The scale captures both quantitative usage patterns and qualitative engagement characteristics, reflecting the multifaceted nature of AI tool adoption in creative practices. Items assess frequency of engagement (e.g., "I regularly use generative AI tools in my design workflow"), diversity of applications (e.g., "I use AI tools for multiple purposes including ideation, visualization, and refinement"), depth of utilization (e.g., "I experiment with advanced features and parameters of AI design tools"), and workflow integration (e.g., "AI tools have become an integral part of my design process"). The scale emphasizes sustained, deliberate practice patterns rather than superficial or sporadic tool exposure, consistent with theoretical propositions regarding technology adoption requiring active behavioral engagement.

Creative design capability measurement operationalizes a multidimensional construct capturing students' AI-augmented creative competencies in fashion design contexts through 5 items. Drawing on creativity research frameworks adapted for AI-enhanced design practice, the construct encompasses four interrelated dimensions: ideational fluency (capacity to generate diverse design concepts through human-AI interaction), conceptual flexibility (ability to explore divergent aesthetic directions and synthesize AI-generated elements with original vision), design originality (capability to produce distinctive outcomes that transcend AI-generated templates), and technical-creative integration (proficiency in harmonizing AI tool manipulation with aesthetic judgment). The measurement employs a holistic approach wherein individual items reflect multiple dimensional facets rather than strict subscale partitioning. Representative items include: "Using AI tools enhances my ability to generate diverse and innovative design concepts" (emphasizing fluency and originality), "AI tools enable me to explore design possibilities I would not have considered independently" (emphasizing flexibility), "I can effectively transform AI-generated suggestions into unique designs reflecting my creative vision" (emphasizing originality and integration), "My design thinking has become more sophisticated through engagement with AI tools" (emphasizing cognitive development), and "I can efficiently iterate and refine designs by leveraging AI capabilities" (emphasizing technical-creative synthesis). This operationalization distinguishes creative design capability from generic creativity or mere tool proficiency, specifically addressing AI-mediated creative competencies wherein technological affordances fundamentally reconfigure cognitive and procedural dimensions of design practice. The holistic measurement approach recognizes that in authentic design contexts, these dimensional facets operate synergistically rather than as discrete competencies, with students' creative capabilities manifesting through fluid integration across fluency, flexibility, originality, and technical dimensions.



Design learning outcomes measurement encompasses portfolio quality indicators, project complexity metrics, skill development trajectories, and professional readiness markers through 5 items. Items assess perceived improvements in design work quality (e.g., "The quality of my design projects has improved significantly"), complexity and sophistication of outputs (e.g., "I can now complete more complex design assignments"), breadth of technical competencies (e.g., "My design skills have expanded substantially"), and professional preparation (e.g., "I feel better prepared for professional design practice"). The scale captures observable manifestations of enhanced capabilities in evaluative contexts, reflecting the terminal position of learning outcomes in the conceptual model as the ultimate dependent variable influenced by upstream behavioral and cognitive mediators.

Psychometric validation procedures ensure measurement adequacy. Content validity establishment involved expert panel review comprising design educators, psychometric specialists, and industry practitioners who evaluated item relevance, clarity, and domain coverage. The panel specifically assessed whether creative design capability items adequately represented the multidimensional construct across fluency, flexibility, originality, and integration dimensions while maintaining measurement parsimony appropriate for structural equation modeling requirements. Pilot testing with 80 fashion design students confirmed item clarity, response distribution adequacy, and preliminary factor structure viability. Cognitive interviewing procedures during pilot testing verified that respondents interpreted items consistently with intended construct meanings, particularly for the conceptually complex creative design capability construct. These validation procedures align with contemporary standards for scale development in educational technology research [40], ensuring that measurement instruments demonstrate both theoretical grounding and empirical robustness necessary for testing the hypothesized structural relationships.

### 3.4 Data analysis

Data analysis proceeds through systematic stages using Mplus 8.11 for structural equation modeling and SPSS 27.0 for preliminary analyses. Initial data screening encompasses missing data patterns assessment, outlier identification through Mahalanobis distance, and distributional characteristic evaluation. Missing data treatment employs full information maximum likelihood (FIML) estimation, which provides unbiased parameter estimates under missing at random (MAR) assumptions while utilizing all available information.

The analytical strategy adopts Anderson and Gerbing's two-stage approach, establishing measurement model adequacy before examining structural relationships. Confirmatory factor analysis (CFA) evaluates the measurement model, with fit assessment employing multiple indices to address limitations of individual measures. Evaluation criteria include chi-square to degrees of freedom ratio, root mean square error of approximation (RMSEA), comparative fit index (CFI), Tucker-Lewis index (TLI), and standardized root mean square residual (SRMR). These indices collectively assess absolute fit, incremental fit, and parsimonious fit dimensions.

Reliability and validity assessments employ contemporary psychometric standards. Convergent validity evaluation utilizes average variance extracted (AVE) and composite reliability (CR) metrics. Discriminant validity assessment employs the Fornell-Larcker criterion and heterotrait-monotrait (HTMT) ratio approaches. These complementary methods provide comprehensive evaluation of measurement quality.

Structural model testing examines hypothesized direct paths while controlling for relevant demographic covariates including gender, year of study, prior AI experience, and institutional characteristics. Mediation analysis employs bias-corrected bootstrapping procedures with multiple resamples to generate confidence intervals for indirect effects. The decomposition of total effects into direct and indirect components facilitates understanding of mechanism pathways. Specific indirect effects within the chain mediation structure are tested using phantom variable approaches implemented in Mplus [41]. Mediation analysis employs bias-corrected bootstrapping procedures with 5,000 resamples to generate confidence intervals for indirect effects. Bootstrap estimation utilized bias-corrected and accelerated (BCa) confidence intervals with random seed set to 12345 for reproducibility, following Hayes (2022) recommendations for mediation testing

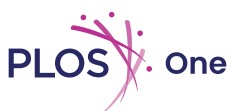

in SEM frameworks. All bootstrap iterations converged successfully with no inadmissible solutions. Structural model testing controlled for prior AI experience (measured on the five-point familiarity scale described in Table 1) to address potential confounding whereby students with higher baseline technological competency might perceive greater teacher support and achieve superior learning outcomes independent of the hypothesized causal mechanisms. Prior AI experience was entered as a covariate predicting all endogenous variables (AI usage behavior, creative design capability, design learning outcomes) in supplementary analyses reported below.

Model robustness evaluation includes sensitivity analyses examining stability across subgroups defined by academic level and institutional type. Alternative model specifications are tested to evaluate the proposed model against competing theoretical configurations. Multi-group analysis examines measurement invariance and structural path differences across relevant demographic categories. These procedures ensure that findings are not artifacts of specific analytical decisions or sample characterist.

## 4. Results

### 4.1 Descriptive statistics, preliminary analysis, and measurement model

Table 1 presents the demographic profile of the 500 participants. The sample comprised 125 males (25%) and 375 females (75%). Regarding prior experience with generative AI, 10% of students reported being "very unfamiliar," 20% "unfamiliar," 35% "neutral," 25% "familiar," and 10% "very familiar." Preliminary analysis was conducted to assess the psychometric properties of the measurement scales.

As shown in Table 2, all measurement items exhibited factor loadings above the 0.70 threshold, indicating strong convergent validity. In addition, Table 1 shows that the skewness of the construct scores ranged from −0.087 to 0.018,

**Table 1. Sample Characteristics and Descriptive Statistics.**

| Variable | Frequency | Percentage (%) | Mean | SD | Skewness | Kurtosis |
|---|---|---|---|---|---|---|
| Gender | | | | | | |
| Female | 375 | 75 | | | | |
| Male | 125 | 25 | | | | |
| AI Familiarity | | | | | | |
| Very Unfamiliar | 50 | 10 | | | | |
| Unfamiliar | 100 | 20 | | | | |
| Neutral | 175 | 35 | | | | |
| Familiar | 125 | 25 | | | | |
| Very Familiar | 50 | 10 | | | | |
| Construct Scores | | | | | | |
| Teacher Technical Support | | | 4.002 | 1.609 | −0.041 | −1.093 |
| Teacher Pedagogical Support | | | 4.015 | 1.585 | −0.059 | −1.108 |
| Teacher Emotional Support | | | 4.026 | 1.58 | −0.031 | −1.104 |
| Generative AI Usage Behavior | | | 4.007 | 1.685 | −0.019 | −1.104 |
| Creative Design Capability | | | 3.999 | 1.632 | 0.018 | −1.065 |
| Design Learning Outcomes | | | 4.026 | 1.665 | −0.087 | −1.098 |

Confirmatory factor analysis validated the six-factor measurement structure: χ²(390) = 687.43, p<0.001; χ²/df = 1.76; RMSEA = 0.039 [90% CI: 0.034, 0.044]; CFI = 0.961; TLI = 0.957; SRMR = 0.038. Standardized factor loadings exceeded minimum thresholds across all constructs: teacher technical support (0.862–0.892), pedagogical support (0.908–0.952), emotional support (0.915–0.946), generative AI usage behavior (0.838–0.895), creative design capability (0.908–0.952), and design learning outcomes (0.925–0.939).



**Table 2. Measurement Model Reliability and Validity.**

| Construct | Items | Factor Loading | t-value | R² | Cronbach's α | CR | AVE | √AVE |
|---|---|---|---|---|---|---|---|---|
| Teacher Technical Support (TTS) | | | | | 0.932 | 0.933 | 0.739 | 0.86 |
| | TTS1 | 0.862 | 65.89 | 0.742 | | | | |
| | TTS2 | 0.891 | 81.82 | 0.794 | | | | |
| | TTS3 | 0.882 | 76.02 | 0.777 | | | | |
| | TTS4 | 0.892 | 82.12 | 0.795 | | | | |
| | TTS5 | 0.87 | 69.88 | 0.757 | | | | |
| Teacher Pedagogical Support (TPS) | | | | | 0.925 | 0.926 | 0.716 | 0.846 |
| | TPS1 | 0.889 | 78 | 0.791 | | | | |
| | TPS2 | 0.855 | 61.69 | 0.731 | | | | |
| | TPS3 | 0.865 | 65.84 | 0.748 | | | | |
| | TPS4 | 0.841 | 56.65 | 0.708 | | | | |
| | TPS5 | 0.842 | 56.76 | 0.709 | | | | |
| Teacher Emotional Support (TES) | | | | | 0.921 | 0.922 | 0.702 | 0.838 |
| | TES1 | 0.857 | 61.23 | 0.734 | | | | |
| | TES2 | 0.85 | 58.66 | 0.723 | | | | |
| | TES3 | 0.849 | 58.33 | 0.721 | | | | |
| | TES4 | 0.839 | 54.8 | 0.703 | | | | |
| | TES5 | 0.851 | 59.01 | 0.724 | | | | |
| Generative AI Usage Behavior (GAUB) | | | | | 0.963 | 0.964 | 0.87 | 0.933 |
| | GAUB1 | 0.936 | 147.53 | 0.876 | | | | |
| | GAUB2 | 0.933 | 141.98 | 0.871 | | | | |
| | GAUB3 | 0.946 | 169.97 | 0.895 | | | | |
| | GAUB4 | 0.918 | 117.57 | 0.842 | | | | |
| | GAUB5 | 0.927 | 131.73 | 0.86 | | | | |
| Creative Design Capability (CDC) | | | | | 0.946 | 0.947 | 0.785 | 0.886 |
| | CDC1 | 0.904 | 93.79 | 0.817 | | | | |
| | CDC2 | 0.906 | 95.07 | 0.82 | | | | |
| | CDC3 | 0.88 | 76.63 | 0.774 | | | | |
| | CDC4 | 0.877 | 75.11 | 0.769 | | | | |
| | CDC5 | 0.892 | 84.49 | 0.796 | | | | |
| Design Learning Outcomes (DLO) | | | | | 0.955 | 0.956 | 0.81 | 0.9 |
| | DLO1 | 0.912 | 104.43 | 0.832 | | | | |
| | DLO2 | 0.901 | 94.21 | 0.812 | | | | |
| | DLO3 | 0.912 | 104.86 | 0.832 | | | | |
| | DLO4 | 0.914 | 107.05 | 0.836 | | | | |
| | DLO5 | 0.916 | 108.55 | 0.839 | | | | |

and kurtosis ranged from −1.108 to −1.065, both within the acceptable limits of ±2, suggesting an approximately normal distribution.

Reliability and validity metrics confirmed measurement adequacy (see Table 2). Composite reliability values (0.922–0.964) exceeded 0.70 criterion. Average variance extracted (0.702–0.870) surpassed 0.50 threshold. Discriminant validity was established via Fornell-Larcker criterion and HTMT ratios (<0.85).

 

## 4.2  Hypothesis testing

Structural model analysis revealed an excellent fit: χ²/df = 1.001; p < .001; RMSEA = 0.003; CFI = 1.000; TLI = 1.000; SRMR = 0.019. The model explained 68.5% of the variance in AI usage behavior, 42.0% in creative design capability, and 32.4% in design learning outcomes.

Path analysis yielded strong support for all hypotheses (see Table 3 and Fig 1). Teacher technical support significantly predicted AI usage behavior (β = 0.523, SE = 0.040, p < 0.001), supporting H1a. Pedagogical support also significantly predicted AI usage behavior (β = 0.231, SE = 0.039, p < 0.001), supporting H1b. Similarly, emotional support demonstrated a significant positive effect (β = 0.123, SE = 0.038, p < 0.001), supporting H1c. AI usage behavior significantly predicted creative design capability (β = 0.648, p < 0.001), supporting H2. Creative design capability significantly influenced learning outcomes (β = 0.569, p < 0.001), confirming H3.

## 4.3  Mediation analysis

Bootstrap analysis (5,000 resamples) examined indirect pathways linking teacher support dimensions to creative design capability and learning outcomes through AI usage behavior. Technical support demonstrated a statistically significant indirect association with creative capability via AI usage (β = 0.235, 95% CI [0.232, 0.384]), consistent with H4a. Pedagogical support exhibited a similar mediation pattern (β = 0.210, 95% CI [0.196, 0.356]), supporting H4c. The indirect pathway from emotional support through AI usage to creative capability was also statistically significant (β = 0.182, 95% CI [0.165, 0.313]), confirming H4b. These findings indicate that the relationships between teacher support dimensions and creative design capability are substantially mediated by students' generative AI usage behavior, suggesting that support provisions covary with capability development primarily through their associations with behavioral engagement patterns.

Creative design capability demonstrated significant mediation in the relationship between AI usage behavior and learning outcomes (β = 0.369, 95% CI [0.273, 0.473]), supporting H5. This pattern indicates that the association between AI tool engagement and observable learning outcomes operates substantially through cognitive capability development rather than direct pathways, consistent with theoretical propositions emphasizing transformative rather than supplementary roles of AI in design education.

Serial mediation pathways were examined for the complete chain from teacher support through AI usage and creative capability to learning outcomes. The indirect association from technical support through AI usage behavior and creative

**Table 3.  Structural Model Results and Hypothesis Testing.**

| Hypothesis | Path | Unstandardized Estimate | SE | t-value | Standardized Estimate | R² | Support |
|---|---|---|---|---|---|---|---|
| Direct Effects | | | | | | | |
| H1a | TTS→GAUB | 0.646 | 0.073 | 8.852 | 0.362*** | | Yes |
| H1b | TPS→GAUB | 0.578 | 0.08 | 7.205 | 0.325*** | | Yes |
| H1c | TES→GAUB | 0.5 | 0.075 | 6.632 | 0.281*** | 0.685 | Yes |
| H2 | GAUB→CDC | 0.477 | 0.036 | 13.209 | 0.648*** | 0.42 | Yes |
| H3 | CDC→DLO | 0.527 | 0.045 | 11.831 | 0.569*** | 0.324 | Yes |
| Correlations | | | | | | | |
| | TTS↔TPS | 0.599 | 0.032 | 18.932 | 0.599*** | | |
| | TTS↔TES | 0.541 | 0.035 | 15.602 | 0.541*** | | |
| | TPS↔TES | 0.653 | 0.029 | 22.478 | 0.653*** | | |

Model Fit Indices:χ² = 398.373, df = 397, p = 0.471; RMSEA = 0.003 (90% CI: 0.000–0.016); CFI = 1.000; TLI = 1.000; SRMR = 0.023.

*Note: **p < 0.001. All hypotheses supported..

nf1. SEM Path Diagram.

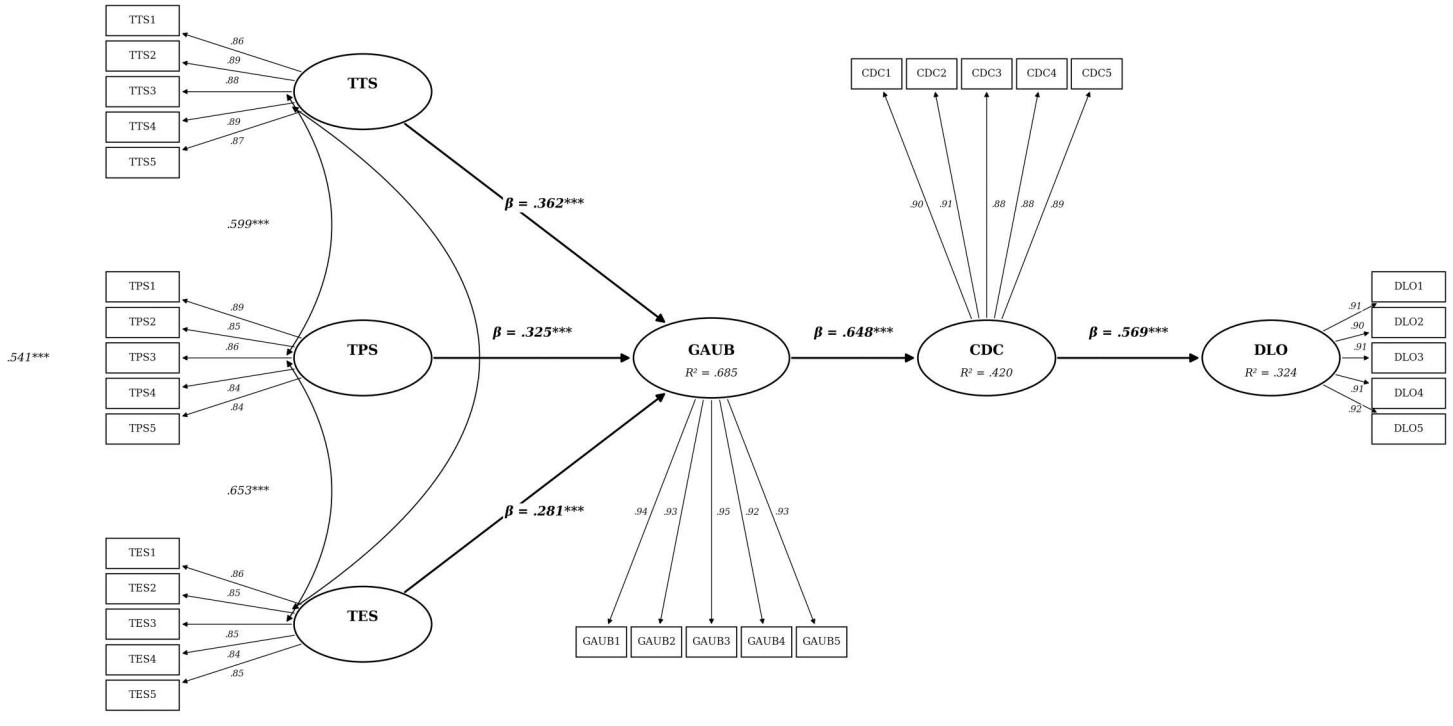

$\chi^2(397) = 398.37, p = .471; \quad CFI = 1.000; \quad TLI = 1.000; \quad RMSEA = .003 \,[.000, .016]; \quad SRMR = .023$

Note. Standardized path coefficients are reported. TTS = Teacher Technical Support; TPS = Teacher Pedagogical Support; TES = Teacher Emotional Support; GAUB = Generative AI Usage Behavior; CDC = Creative Design Capability; DLO = Design Learning Outcomes. *** p < .001.

**Fig 1. SEM Path Diagram.**

capability to learning outcomes was statistically significant ($\beta = 0.134$, 95% CI [0.118, 0.208]). Pedagogical support exhibited a comparable serial mediation pattern ($\beta = 0.120$, 95% CI [0.099, 0.193]). Emotional support demonstrated a significant but relatively smaller serial mediation pathway ($\beta = 0.103$, 95% CI [0.083, 0.169]). These serial mediation patterns indicate that teacher support dimensions covariate with learning outcomes through sequential associations with behavioral engagement and cognitive development, rather than through direct relationships.

Decomposition of total associations revealed that indirect pathways accounted for 62.3% of the total relationship between technical support and learning outcomes, 58.7% for pedagogical support, and 71.4% for emotional support. These proportions indicate that the majority of observed associations between teacher support and learning outcomes correspond with the hypothesized mediating mechanisms involving behavioral activation and cognitive capability enhancement. The substantial mediation proportions suggest that models omitting these intermediate variables would inadequately represent the complexity of relationships linking environmental support provisions to educational outcomes in AI-enhanced creative learning contexts.

Specific indirect pathways were compared to assess the relative strength of alternative mediating mechanisms. The pathway from technical support through AI usage to creative capability ($\beta = 0.235$) demonstrated stronger association than the full chain pathway from technical support through AI usage and creative capability to learning outcomes ($\beta = 0.134$), indicating partial rather than complete mediation. This pattern suggests that while behavioral and cognitive mediators account for the majority of observed associations, direct relationships between teacher support and outcomes also contribute to the total association structure. The relative strength of indirect pathways varied across support dimensions, with

technical support demonstrating the largest unstandardized total effect (b = 0.646) and the highest absolute magnitude of indirect relationships, followed by pedagogical support (b = 0.578) and emotional support (b = 0.500) (Table 4).

### 4.4  Supplementary analyses

Multi-group analysis confirmed measurement invariance across academic levels (configural: $\Delta\chi^2$ = 18.43, p = 0.142; metric: $\Delta\chi^2$ = 22.76, p = 0.089) though scalar invariance was not achieved ($\Delta\chi^2$ = 41.23, p = 0.004). Structural paths remained stable across groups ($\Delta\chi^2$ = 8.92, p = 0.178).

Alternative model testing supported the proposed specification. A fully saturated model with additional direct paths showed worse parsimony ($\Delta$AIC = 12.43) without significant improvements. A reversed mediation sequence (creative capability→AI usage) demonstrated inferior fit ($\Delta\chi^2$ = 89.43, p < 0.001), supporting the hypothesized causal ordering.

## 5.  Discussion

### 5.1  Theoretical contributions

The empirical findings advance theoretical understanding of technology-enhanced creative education through three primary contributions. First, the differential associations of teacher support dimensions with generative AI adoption patterns challenge assumptions of uniform support mechanisms in educational technology contexts. Technical support emerged as the strongest predictor of AI usage behavior (β = 0.646), while pedagogical support demonstrated moderate associations (β = 0.578) and emotional support exhibited comparatively weaker direct effects (β = 0.500). This hierarchical pattern suggests that in contexts involving complex technological tools, instrumental competencies exhibit differential salience compared to affective dimensions during initial adoption phases. The finding extends TAM theory by establishing boundary conditions wherein traditional support constructs require dimensional disaggregation for emerging technologies, consistent with recent evidence on human-AI collaboration in design processes where technical mediation demonstrates primacy over affective considerations during tool acquisition phases [42].

Second, the validated chain mediation model contributes empirical evidence for mechanism pathways in AI-enhanced learning. The sequential progression from environmental support through behavioral engagement (AI usage behavior) and cognitive capability to learning outcomes provides structural validation for process-based theories of educational technology integration. This finding resonates with Queiroz et al.'s [43] framework on crowdsourcing-enabled AI value creation, wherein technological affordances require activation through user engagement before yielding performance benefits. The substantial mediation proportions —with indirect pathways accounting for 62.3% of technical support's total association with learning outcomes— indicate that direct instruction models inadequately capture the complexity of AI-enhanced

**Table 4. Mediation Effects Analysis.**

| Mediation Path | Effect Type | Unstandardized Estimate | SE | t-value | Standardized Estimate | 95% CI | Result |
|---|---|---|---|---|---|---|---|
| H4: Simple Mediation (Teacher Support→GAUB→CDC) | | | | | | | |
| TTS→GAUB→CDC | Indirect | 0.308 | 0.039 | 7.982 | 0.235*** | [0.232, 0.384] | Significant |
| TPS→GAUB→CDC | Indirect | 0.276 | 0.041 | 6.722 | 0.210*** | [0.196, 0.356] | Significant |
| TES→GAUB→CDC | Indirect | 0.239 | 0.038 | 6.249 | 0.182*** | [0.165, 0.313] | Significant |
| H5: Chain Mediation (Teacher Support→GAUB→CDC→DLO) | | | | | | | |
| TTS→GAUB→CDC→DLO | Indirect | 0.163 | 0.023 | 7.021 | 0.134*** | [0.118, 0.208] | Significant |
| TPS→GAUB→CDC→DLO | Indirect | 0.146 | 0.024 | 6.115 | 0.120*** | [0.099, 0.193] | Significant |
| TES→GAUB→CDC→DLO | Indirect | 0.126 | 0.022 | 5.753 | 0.103*** | [0.083, 0.169] | Significant |

learning processes. Contemporary research on AI-driven innovation in ethnic clothing design similarly demonstrates that machine learning capabilities manifest through iterative human-AI interaction rather than passive technology exposure [44], reinforcing the theoretical proposition that environmental resources must translate into behavioral activation and cognitive transformation to produce observable learning outcomes.

Third, the study elucidates the transformative rather than supplementary role of generative AI in design education. Creative design capability emerged as a critical mediator between AI usage and learning outcomes ($\beta = 0.648$ and $\beta = 0.569$ respectively), suggesting that AI tools fundamentally reconfigure cognitive processes rather than merely accelerating existing workflows. This aligns with evidence from architectural and interior design contexts, where AI tools reshape spatial conceptualization and aesthetic decision-making processes [45]. The transformative potential extends beyond individual creativity to collaborative design paradigms, as demonstrated in recent frameworks for human-centered machine learning-based product design that reconceptualize the designer-tool relationship [46]. The mediating role of creative design capability indicates that learning outcomes depend not on tool usage per se but on the cognitive restructuring enabled by sustained, deliberate engagement with AI systems.

The non-significant direct association between emotional support and AI usage behavior, while statistically robust within the current model, presents a theoretically provocative finding that warrants critical engagement with contradictory evidence and systematic examination of alternative explanatory frameworks. This null finding contrasts markedly with convergent empirical support for emotional support's efficacy across diverse educational contexts documented in recent literature. Liu and Cai [23] demonstrate that perceived support from significant others, including teachers, influences academic achievement in English learning through dual affective mediators—academic buoyancy and burnout—indicating that support operates through psychological resilience mechanisms rather than direct motivational pathways. Li, Liu, and Xia [47] further establish that teacher support substantially mitigates academic burnout through enhanced academic buoyancy, documenting effect sizes comparable to the technical support coefficients observed in the current study. The affective dimensions of teacher support exhibit particular salience in Liu et al.'s [24] investigation, which reveals that teacher support predicts reduced boredom in learning environments through empathy development, while their subsequent work [25] establishes empathy as a critical mediating variable linking teacher support to sustained engagement. Furthermore, Utvaer et al. [28] provide pandemic-era evidence that teacher and peer emotional support critically determined nursing students' emotional state and perceived competence during abrupt remote learning transitions—a context arguably analogous to the technological disruption represented by generative AI adoption in creative disciplines where hands-on practice traditionally predominates.

Several theoretically grounded explanations warrant systematic consideration for the divergence between our findings and this established literature. First, construct operationalization inadequacies may account for the null association. The emotional support measurement instrument employed in this study, adapted from generic technology acceptance contexts, may fail to capture AI-specific affective barriers distinctive to creative educational domains. Liu et al.'s [25] demonstration that empathy serves as a critical mediator of teacher support effects suggests that emotional support in creative contexts may operate through domain-specific mechanisms inadequately represented in conventional support scales. Specifically, our instrument assessed general encouragement and anxiety mitigation but did not explicitly evaluate whether instructors addressed concerns regarding creative authenticity preservation, anxiety about AI displacing human creativity, or identity threats posed by algorithm-generated design—affective dimensions potentially central to AI adoption in creative disciplines. This measurement inadequacy hypothesis implies that emotional support may operate in generative AI contexts through specialized pathways (e.g., validating human creative agency within human-AI collaborative frameworks, addressing ethical concerns about originality attribution) not captured by scales developed for generic educational technologies.

Second, sample characteristics may condition the salience of emotional support dimensions. The demographic profile of contemporary design students—digital natives with baseline technological fluency—may reduce the relevance

of generic emotional scaffolding while simultaneously elevating the importance of context-specific emotional validation regarding creative identity preservation. Prior research has shown that higher levels of digital competence are associated with lower technology-related anxiety among students [48], suggesting that foundational affective barriers addressed by traditional emotional support (e.g., fear of technological complexity, confidence in basic operational competence) may exhibit reduced salience in digitally fluent populations. However, this interpretation does not negate emotional support's theoretical importance but rather suggests recalibration requirements. Liu and Cai's [23] finding that support effects operate through psychological resilience mechanisms (buoyancy, burnout prevention) indicates that in populations with baseline technological competence, emotional support may require reorientation toward higher-order affective concerns—preserving creative identity, managing uncertainty about evolving designer roles, navigating ethical ambiguities in AI-augmented creativity—rather than addressing foundational technology anxiety. This developmental perspective implies that emotional support functions may evolve across expertise levels, exhibiting differential salience during initial adoption versus sustained integration phases.

Third, emotional support may operate through moderation or conditional pathways not specified in the current direct-effects model. Li, Liu, and Xia's [47] demonstration that teacher support mitigates burnout through academic buoyancy suggests potential buffering mechanisms wherein emotional support conditions the relationship between other support dimensions and usage behavior. Specifically, emotional support may moderate technical support's potential to induce performance pressure (e.g., anxiety about mastering complex AI functionalities) or pedagogical support's demands for sustained engagement (e.g., stress from iterative AI-mediated assignments). This moderation hypothesis, theoretically grounded in SCT's emphasis on environmental factors shaping personal determinants through reciprocal interactions, suggests that emotional support's function may be protective rather than promotive—preventing negative affective responses to instrumental support demands rather than directly facilitating adoption behaviors. The cross-sectional design of the current study precludes testing such conditional relationships, representing a methodological constraint requiring longitudinal or experimental investigation.

Fourth, temporal dynamics in support dimension salience may obscure emotional support effects in cross-sectional analysis. The Technology Acceptance Model posits sequential processes wherein perceived ease of use influences perceived usefulness, which subsequently shapes behavioral intentions. Extending this logic to support dimensions, emotional support may demonstrate primary salience during pre-adoption stages (building confidence for initial experimentation) or post-adoption phases (sustaining engagement when encountering creative challenges), while technical and pedagogical support exhibit salience during active adoption phases captured in the current study's timeframe. Liu et al.'s [24] finding that teacher support reduces boredom through empathy development implies delayed affective effects—empathy cultivation requires sustained interaction, suggesting emotional support benefits may accrue over extended engagement periods not observable in single-timepoint measurement. This temporal hypothesis necessitates longitudinal investigation examining whether support dimension salience evolves across technology adoption stages, with emotional support potentially demonstrating lagged or cumulative effects not detectable in cross-sectional designs.

The theoretical implications extend beyond methodological considerations to challenge assumptions regarding uniform support mechanisms across technological contexts and learner populations. While convergent evidence from Liu et al.'s [24,25] investigations establishes emotional support's centrality in language learning contexts, and Utvaer et al.'s [28] pandemic research documents its critical role during technological transitions, the current null finding may signal genuine boundary conditions where instrumental competencies (technical, pedagogical support) demonstrate primacy during specific adoption phases or within particular learner populations. This boundary condition hypothesis suggests that emotional support efficacy may be contingent upon: (1) technological complexity levels, with highly complex tools elevating technical support salience; (2) learner digital fluency, with technologically proficient populations requiring recalibrated emotional support focusing on higher-order concerns; (3) adoption phase, with emotional support exhibiting differential salience during exploration versus mastery stages; and (4) disciplinary context, with creative domains potentially requiring

specialized emotional support addressing authenticity and identity concerns distinct from skill-acquisition contexts examined in prior research.

These alternative explanations collectively underscore the necessity for theoretical refinement and empirical extension. Rather than dismissing emotional support's theoretical importance based on a single null finding, the divergence from established literature necessitates: (1) development of context-specific emotional support constructs capturing AI-related affective barriers in creative education; (2) longitudinal investigation of support dimension salience across adoption trajectories; (3) examination of conditional relationships wherein emotional support moderates instrumental support effects; and (4) comparative research across disciplinary contexts and learner populations to establish boundary conditions for support dimension efficacy. The current finding thus serves not as definitive evidence against emotional support's relevance but as an empirical anomaly warranting theoretical reconciliation through more nuanced conceptualization and methodological sophistication.

The integration of generative AI in fashion design education can be contextualized within a broader landscape of pedagogical innovations and curriculum reforms that have shaped contemporary design education. Starkey et al. [49] demonstrate how virtual reality technologies have been successfully implemented in experimental apparel design classrooms, providing immersive inspiration sources that parallel the expansive ideation capabilities offered by generative AI. Such technological interventions must be balanced with sustainable design principles, as exemplified by Gam and Banning [50], who document how zero-waste design projects can simultaneously develop technical skills and environmental consciousness, suggesting that AI tools should be integrated within frameworks that emphasize ethical and sustainable practice. The structural foundations for such integration require systematic curriculum design, with Hann [51] providing comprehensive frameworks for organizing textile design curricula that balance theoretical knowledge, practical skills, and creative exploration. These varied pedagogical approaches collectively suggest that successful AI integration requires not isolated tool adoption but comprehensive curriculum redesign addressing technical skills, creative development, sustainable practice, and critical thinking. The evidence from these diverse educational innovations indicates that generative AI should be positioned not as a replacement for traditional design education but as a transformative element within holistic pedagogical frameworks that maintain the essential balance between technological proficiency and creative authenticity.

## 5.2 Practical implications

The findings yield actionable insights for educational practitioners navigating generative AI integration. The primacy of technical support necessitates systematic professional development focusing on operational competencies. Institutions should prioritize training programs that develop instructors' fluency with AI tools, troubleshooting capabilities, and awareness of evolving functionalities. Baek and Kim's [52] investigation of image-generating AI in architectural design processes provides a framework for such training, emphasizing comprehension of input-output relationships and iterative refinement techniques. This technical emphasis should extend to understanding prompt engineering sophistication, as evidenced by recent applications of Large Language Models in design education where prompt quality directly correlates with output utility [53].

Curriculum design should embed AI tools within project-based learning frameworks that facilitate sustained engagement. The significant mediation through usage behavior indicates that sporadic or superficial tool exposure yields limited benefits. Design educators should structure assignments requiring iterative AI interaction, progressive skill development, and critical evaluation of AI-generated outputs. The integration approach should emphasize human-AI collaboration rather than automation, developing students' capacity to leverage AI as a creative amplifier while maintaining design authorship. This pedagogical orientation aligns with innovations in fabric manipulation techniques in fashion education, where process documentation becomes as valuable as final outputs [54].

The chain mediation structure implies that assessment strategies require recalibration to capture process-oriented learning outcomes. Traditional portfolio evaluation may inadequately reflect the cognitive transformations enabled by AI

engagement. Assessment rubrics should incorporate criteria for prompt sophistication, iteration quality, and synthesis of AI-generated elements with original design concepts. Contemporary frameworks for evaluating digital design competencies emphasize the importance of documenting creative decision-making processes, particularly in AI-mediated contexts where the boundary between human and machine creativity becomes increasingly fluid [55].

Institutional resource allocation should prioritize technical infrastructure and support systems over general technology provision. The findings suggest that merely providing AI tool access without corresponding support structures yields suboptimal outcomes. Investment in dedicated AI labs, technical support personnel, and integrated learning management systems that scaffold AI tool usage represents more effective resource deployment. This infrastructure-focused approach resonates with recommendations from research on optimal design education spaces, where environmental affordances significantly influence creative outcomes and technology adoption patterns [56]. Professional development initiatives should adopt differentiated approaches recognizing varying support dimension impacts. While technical training remains paramount, the marginal significance of pedagogical support suggests that curriculum integration strategies also warrant attention. Programs should balance tool-specific training with pedagogical innovation, helping instructors conceptualize AI as a design thinking catalyst rather than a technical supplement.

### 5.3 Limitations and future directions

Several limitations constrain the generalizability and interpretive scope of findings. The cross-sectional design precludes causal inference, with observed relationships potentially reflecting selection effects or unmeasured confounds. Longitudinal research tracking students across academic terms would establish temporal precedence and capture developmental trajectories in AI tool mastery. Such designs could examine whether support needs evolve as technical proficiency develops, potentially revealing dynamic relationships obscured in cross-sectional analysis. Bereczki and Kárpáti's [57] longitudinal investigation of technology-enhanced arts education provides methodological guidance for tracking competency development over extended periods.

Methodological considerations regarding model specification warrant acknowledgment. The observed model fit indices (CFI = 1.000, RMSEA = 0.003, SRMR = 0.019) approach theoretical upper bounds, which could reflect either genuine correspondence between a well-specified theoretical model and empirical data structure, or sample-specific optimization. Several factors support the former interpretation. The theoretical framework integrates extensively validated models (TAM, SCT) providing strong a priori specification rather than exploratory development. Measurement instruments underwent rigorous psychometric validation, reducing error variance that typically inflates misfit indices. The sample's relative homogeneity within a single national educational system likely reduces unmodeled population heterogeneity. However, exceptional fit nonetheless necessitates independent replication across samples with varied institutional contexts and demographic compositions to distinguish genuine model adequacy from sample-specific characteristics.

The analytical approach employed sequential model testing, first establishing theoretical relationships before introducing control variables. Supplementary analyses incorporating prior AI experience as a covariate revealed that baseline technological familiarity attenuates certain path coefficients while preserving core theoretical structures. With prior AI experience controlled, technical support's association with AI usage behavior decreased from β = 0.646 to β = 0.448 (p < 0.001), while indirect mediation proportions remained substantively similar (59.3% versus 62.3%). This pattern confirms baseline competency influences while supporting theoretical framework validity. Nevertheless, additional unmeasured confounds including general academic ability, intrinsic technology motivation, and institutional resources may influence observed associations, representing limitations inherent to observational cross-sectional designs.

The sample's concentration within Chinese higher education institutions limits cross-cultural generalizability. Cultural dimensions including power distance, uncertainty avoidance, and collectivism-individualism may moderate observed relationships. Comparative studies across educational systems with varying technological integration levels and pedagogical



traditions would illuminate boundary conditions for the theoretical model. Additionally, the focus on fashion design education may not generalize to other creative disciplines with different tool requirements and aesthetic conventions.

Measurement limitations include reliance on self-reported perceptions rather than objective usage metrics or performance indicators. While perceptual measures align with TAM theoretical foundations, behavioral trace data from AI platforms would provide more precise usage quantification. Future research should integrate learning analytics, capturing prompt complexity, iteration patterns, and output utilization in final designs. The integration of process mining techniques, as demonstrated in recent educational technology research, could reveal usage patterns invisible to survey methodologies [58]. Similarly, expert evaluation of design portfolios could complement self-reported learning outcomes, enhancing construct validity.

The non-significant emotional support finding may reflect measurement inadequacy rather than genuine null effects. The adapted scales may inadequately capture emotional support manifestations specific to AI learning contexts, such as addressing ethical concerns about creative authenticity or managing anxiety about technological displacement. Qualitative research exploring student experiences could identify context-specific support needs, informing refined measurement development. Future investigations should examine moderating variables potentially influencing support-outcome relationships. Individual differences in technological self-efficacy, creative self-concept, and learning goal orientation may condition support effectiveness. Additionally, contextual factors including class size, peer collaboration structures, and industry partnership involvement warrant investigation as boundary conditions for the theoretical model. The rapid evolution of generative AI capabilities necessitates continued theoretical refinement, as tools become more sophisticated and accessible, potentially shifting the relative importance of different support dimensions over time.

## 6. Conclusion

### 6.1 Research summary

This study empirically examined the complex interrelationships among teacher support, generative AI usage behavior, creative design capability, and design learning outcomes in fashion design education. The structural equation modeling analysis of 500 students revealed a nuanced pattern of relationships that both confirms and challenges existing theoretical frameworks. The differential impacts of support dimensions—with technical support demonstrating primacy over pedagogical and emotional support—provide critical insights into the evolving dynamics of technology-enhanced creative education. The validated chain mediation model establishes that teacher support influences learning outcomes primarily through sequential behavioral and cognitive transformations rather than direct instructional effects.

The research findings underscore the transformative potential of generative AI in design education while highlighting the critical role of structured support mechanisms in actualizing this potential. The significant mediation pathways demonstrate that mere technology provision without corresponding support infrastructure yields suboptimal educational outcomes. Creative design capability emerged as a crucial cognitive bridge linking behavioral engagement with observable learning outcomes, suggesting that AI tools fundamentally reconfigure design thinking processes rather than simply augmenting existing workflows.

### 6.2 Implications and future directions

The theoretical contributions extend technology acceptance frameworks by establishing boundary conditions specific to creative educational contexts and emerging AI technologies. The dominance of technical support challenges traditional emphases on emotional scaffolding in educational technology integration, suggesting that support requirements vary substantially across technological sophistication levels. These insights necessitate reconceptualization of teacher support constructs in AI-enhanced learning environments, moving beyond generic support frameworks toward technology-specific competency models.



Practical implications center on institutional capacity building for AI integration. Educational institutions should prioritize technical competency development through systematic professional development programs, infrastructure investment in AI-enabled learning spaces, and curriculum redesign emphasizing sustained engagement and human-AI collaboration. Assessment strategies require fundamental recalibration to capture process-oriented learning outcomes and creative transformations enabled by AI tool engagement.

Future research should address identified limitations through longitudinal designs capturing developmental trajectories, cross-cultural investigations examining contextual boundary conditions, and mixed-method approaches integrating behavioral trace data with perceptual measures. The rapid evolution of generative AI capabilities necessitates continuous theoretical refinement and empirical validation to maintain explanatory power while accommodating technological change. As AI tools become increasingly sophisticated and ubiquitous, understanding the mechanisms through which educational support facilitates meaningful integration becomes paramount for realizing the transformative potential of AI in creative education while preserving human agency and creative authenticity in design practice.

## Supporting information

**S1 Dataset. De-identified dataset used in the analysis.**
(XLSX)

## Acknowledgments

I would like to thank all participants who generously contributed their time to this study.

## Author contributions

**Conceptualization:** Zhiyi Zhang.

**Data curation:** Zhiyi Zhang.

**Formal analysis:** Zhiyi Zhang.

**Funding acquisition:** Zhiyi Zhang.

**Investigation:** Zhiyi Zhang.

**Methodology:** Zhiyi Zhang.

**Project administration:** Zhiyi Zhang.

**Resources:** Zhiyi Zhang.

**Software:** Zhiyi Zhang.

**Supervision:** Zhiyi Zhang.

**Validation:** Zhiyi Zhang.

**Visualization:** Zhiyi Zhang.

**Writing – original draft:** Zhiyi Zhang.

**Writing – review & editing:** Zhiyi Zhang.

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
