## [Decision Letter · Decision Letter 0]

26 Nov 2025

PONE-D-25-50100Teacher Support for Generative AI Use and Design Learning Outcomes in Fashion Design Education: A Structural Equation Modeling StudyPLOS ONE

Dear Dr. Zhiyi Zhang,

Thank you for submitting your manuscript to PLOS ONE. After careful consideration, we feel that it has merit but does not fully meet PLOS ONE’s publication criteria as it currently stands. Therefore, we invite you to submit a revised version of the manuscript that addresses the points raised during the review process.

Both reviewers agree that the manuscript addresses an important and timely topic: how teacher support influences generative AI use and design learning outcomes in fashion design education. However, substantial revisions are required before the manuscript can proceed.

1. Theoretical and Conceptual Issues

The integration of the Technology Acceptance Model (TAM) and Social Cognitive Theory is incomplete. Key constructs such as perceived usefulness, self-efficacy, and teacher support are presented but not theoretically connected. Central variables, especially creative design capability, lack clear definitions, sub-dimensions, or justification, which weakens the conceptual framework and measurement validity.

2. Overinterpretation of Cross-Sectional Data

The manuscript uses causal language despite relying solely on cross-sectional survey data. Claims about mediation effects and directional influence should be reframed as associations, and limitations regarding temporal ordering must be clearly acknowledged.

3. Methodological Clarifications Needed

Although SEM is appropriate, several concerns must be addressed: unusually perfect fit indices suggest potential overfitting; prior AI experience is not used as a control variable; and the mediation analysis lacks standard reporting details (e.g., BCa intervals, bootstrap settings). Reviewer 2 also requests clearer sampling procedures, bias control, and confirmation that ethics approval covers all participating institutions.

4. Writing and Contextual Strengthening

Reviewer 1 notes issues with verbosity and unclear terminology. Reviewer 2 recommends adding stronger pedagogical context particularly how fashion design education traditionally fosters creativity and how AIGC reshapes ideation and studio practice.

5. Data Availability Requirements

Reviewer 2 indicates that the manuscript does not yet meet PLOS ONE data-availability standards.

Conclusion

The manuscript shows promise but requires major revision in theory development, construct clarity, methodological transparency, and writing precision. A substantially revised version addressing all reviewer concerns will be needed for further consideration

If applicable, we recommend that you deposit your laboratory protocols in protocols.io to enhance the reproducibility of your results. Protocols.io assigns your protocol its own identifier (DOI) so that it can be cited independently in the future. For instructions see: https://journals.plos.org/plosone/s/submission-guidelines#loc-laboratory-protocols. Additionally, PLOS ONE offers an option for publishing peer-reviewed Lab Protocol articles, which describe protocols hosted on protocols.io. Read more information on sharing protocols at . Additionally, PLOS ONE offers an option for publishing peer-reviewed Lab Protocol articles, which describe protocols hosted on protocols.io. Read more information on sharing protocols at https://plos.org/protocols?utm_medium=editorial-email&utm_source=authorletters&utm_campaign=protocols..

We look forward to receiving your revised manuscript.

Kind regards,

Dokun Iwalewa OIuwajana

Academic Editor

PLOS ONE

Journal Requirements:

3. In the online submission form, you indicated that [Data will be made available through the corresponding author under reasonable circumstances.].

5. We notice that your supplementary tables are included in the manuscript file. Please remove them and upload them with the file type 'Supporting Information'. Please ensure that each Supporting Information file has a legend listed in the manuscript after the references list.

Additional Editor Comments:

Both reviewers agree that the manuscript addresses an important and timely topic: how teacher support influences generative AI use and design learning outcomes in fashion design education. However, substantial revisions are required before the manuscript can proceed.

1. Theoretical and Conceptual Issues

The integration of the Technology Acceptance Model (TAM) and Social Cognitive Theory is incomplete. Key constructs such as perceived usefulness, self-efficacy, and teacher support are presented but not theoretically connected. Central variables, especially creative design capability, lack clear definitions, sub-dimensions, or justification, which weakens the conceptual framework and measurement validity.

2. Overinterpretation of Cross-Sectional Data

The manuscript uses causal language despite relying solely on cross-sectional survey data. Claims about mediation effects and directional influence should be reframed as associations, and limitations regarding temporal ordering must be clearly acknowledged.

3. Methodological Clarifications Needed

Although SEM is appropriate, several concerns must be addressed: unusually perfect fit indices suggest potential overfitting; prior AI experience is not used as a control variable; and the mediation analysis lacks standard reporting details (e.g., BCa intervals, bootstrap settings). Reviewer 2 also requests clearer sampling procedures, bias control, and confirmation that ethics approval covers all participating institutions.

4. Writing and Contextual Strengthening

Reviewer 1 notes issues with verbosity and unclear terminology. Reviewer 2 recommends adding stronger pedagogical context particularly how fashion design education traditionally fosters creativity and how AIGC reshapes ideation and studio practice.

5. Data Availability Requirements

Reviewer 2 indicates that the manuscript does not yet meet PLOS ONE data-availability standards.

Conclusion

The manuscript shows promise but requires major revision in theory development, construct clarity, methodological transparency, and writing precision. A substantially revised version addressing all reviewer concerns will be needed for further consideration

Reviewers' comments:

Reviewer's Responses to Questions

**Comments to the Author**

1. Is the manuscript technically sound, and do the data support the conclusions?

Reviewer #1: Partly

Reviewer #2: Yes

2. Has the statistical analysis been performed appropriately and rigorously? 

Reviewer #1: Yes

Reviewer #2: Yes

3. Have the authors made all data underlying the findings in their manuscript fully available?

Reviewer #1: Yes

Reviewer #2: No

4. Is the manuscript presented in an intelligible fashion and written in standard English?

Reviewer #1: No

Reviewer #2: Yes

5. Review Comments to the Author

Reviewer #1: I have the following comments, and please read my annotated pdf file where includes other important suggestions.

1. Although this paper addresses a timely and significant topic—examining how teacher support influences students’ use of generative AI and its subsequent impact on design learning outcomes in fashion design education—it suffers from several notable weaknesses across theoretical, methodological and linguistic dimensions.

2. Theoretically, while the authors claim to integrate the Technology Acceptance Model (TAM) with Social Cognitive Theory (SCT), the integration remains superficial. Key constructs such as “perceived usefulness” (TAM) and “self-efficacy” (SCT) are listed, but their interaction is never explicitly modelled or discussed. For example, the paper does not explain how an increase in students’ self-efficacy (SCT) might moderate or mediate the relationship between perceived usefulness (TAM) and actual AI use, leaving the theoretical contribution thin.

3. In terms of research questions, the three questions do form a logical chain (teacher support → AI use → creative capability → learning outcomes), yet the central mediator—“creative design capability”—is never clearly defined. No sub-dimensions such as fluency, flexibility or originality are provided, so the reader cannot tell whether the construct refers to ideational quantity, aesthetic innovation, or technical proficiency. This conceptual vagueness blurs the focus of the entire study.

4. The conclusions repeatedly over-interpret cross-sectional data causally. A typical sentence reads, “62.3 % of technical support’s effect on learning outcomes operates through behavioural and cognitive mediators.” With only a single-time-point survey, the authors cannot rule out reverse causality (students who already possess strong creative skills may be more likely to seek out AI tools and to perceive higher teacher support), so causal language is unwarranted.

5. Linguistically, the manuscript is generally intelligible, yet it is riddled with verbose and opaque sentences. A representative example: “The findings challenge monolithic conceptualizations of educational support, demonstrating that instrumental competencies supersede affective dimensions in facilitating AI tool adoption.” The term “monolithic conceptualizations” is never defined; readers cannot tell whether it refers to uni-dimensional scales, uniform teacher-training policies, or something else. Another illustration of poor cohesion appears when the authors try to explain the null effect of emotional support: “Alternatively, emotional support may moderate rather than directly influence adoption behaviors, a possibility warranting future investigation through more nuanced analytical frameworks.” The sentence is speculative, references no data, and functions as a rhetorical bridge rather than a logically grounded next step.

6. Methodologically, the choice of structural-equation modelling is appropriate, but several technical problems undermine confidence in the results. First, fit indices are suspiciously perfect (CFI = 1.000, RMSEA = 0.003, SRMR = 0.019), raising the possibility of over-fitting or capitalisation on chance; no sensitivity analyses (e.g., alternative estimators, parcelled vs. item-level indicators) are reported to rule this out. Second, prior AI experience is measured but is used only as a descriptive statistic; it is not entered as a control variable in the structural model, even though experienced users might both perceive higher teacher support and achieve better learning outcomes, thereby inflating path coefficients. Third, the mediation analysis states that “5 000 bootstrap resamples” were run, yet the paper fails to specify whether bias-corrected and accelerated (BCa) confidence intervals were used, what seed was set, or whether any convergence problems occurred, all of which are standard reporting requirements for bootstrap mediation (Hayes, 2022). These omissions make it difficult to replicate or trust the indirect-effect estimates.

Reviewer #2: The manuscript presents an important and timely study on how teacher support influences generative AI (AIGC) adoption and learning outcomes in fashion design education. The quantitative modeling is rigorous, but several areas require theoretical and contextual enrichment.

First, the paper should begin by describing the fashion design curriculum and pedagogy before AIGC, explaining how creativity was traditionally fostered through studio-based, project-oriented, and craft-focused methods such as sketching, textile manipulation, and critique sessions. This context would clarify how AIGC transforms long-standing creative learning practices.

In the literature review, all constructs should be clearly justified with sources. For instance, explain why teacher support is conceptualized as technical, pedagogical, and emotional—each supported by existing educational technology literature—and discuss how this structure ensures content validity. Likewise, justify the operational definitions of “AI usage behavior,” “creative design capability,” and “design learning outcomes” with previous studies, providing sample items for each construct.

The conceptual model and each path should be more explicitly grounded in theory. Teacher support should be linked to AI usage through the Technology Acceptance Model, usage to creative capability through cognitive engagement, and capability to learning outcomes through Social Cognitive Theory. Cite recent works in fashion and design education showing how AIGC enhances ideation, problem-solving, and creativity through iterative co-design processes.

Clarify how learning outcomes were measured—such as portfolio quality, project complexity, or self-assessed skill gains—and justify these indicators. The population, consisting of 500 students across 18 Chinese universities, should include the estimated total population and sampling error (approximately ±4.2% at a 95% confidence level). The manuscript should also describe procedures to avoid online data bias, including verified institutional recruitment, IP filtering, and anonymized responses.

The ethics section appropriately notes approval from the University of Leeds (FAHC 22-010); ensure this covers all participating institutions. Finally, enrich the SEM results by elaborating on how AIGC use concretely stimulates creative design capability—highlighting behavioral and cognitive transformation rather than just statistical relationships.

Overall, the manuscript offers meaningful contributions but needs stronger pedagogical grounding, construct justification, and interpretation of how AIGC operationally produces creativity. Recommended decision: Major Revision.

6. PLOS authors have the option to publish the peer review history of their article (what does this mean?). If published, this will include your full peer review and any attached files.). If published, this will include your full peer review and any attached files.

.

Reviewer #1: No

Reviewer #2: **Yes:** Eric C K ChengEric C K Cheng

---

## [Author Response · Author response to Decision Letter 1]

24 Jan 2026

Response to the Academic Editor

I sincerely thank you for your careful evaluation of my manuscript and for the clear and constructive summary of the key issues requiring revision. I greatly appreciate the opportunity to submit a substantially revised version. Below, I respond point by point to the editor’s comments and outline how each issue has been addressed in the revised manuscript. Detailed responses to Reviewer #1 and Reviewer #2 are provided separately in the following sections.

1. Theoretical and Conceptual Issues

Editor’s comment:

The integration of the Technology Acceptance Model (TAM) and Social Cognitive Theory is incomplete. Key constructs such as perceived usefulness, self-efficacy, and teacher support are presented but not theoretically connected. Central variables, especially creative design capability, lack clear definitions, sub-dimensions, or justification, which weakens the conceptual framework and measurement validity.

Response:

I have substantially revised the Theoretical Framework and Literature Review sections to strengthen the conceptual integration between TAM and SCT.

Specifically, I now explicitly theorise teacher support as an environmental antecedent that shapes both perceived usefulness (TAM) and self-efficacy (SCT), clarifying how these constructs jointly influence students’ generative AI usage behaviour. The interaction between cognitive belief formation (TAM) and capability beliefs (SCT) is now clearly articulated rather than implicitly assumed.

In addition, the central construct creative design capability has been reconceptualised and clearly defined based on established design creativity and design cognition literature. It is now operationalised through theoretically grounded sub-dimensions (e.g., ideational fluency, problem-solving flexibility, and integrative design capability), with explicit justification for its role as a mediating mechanism between AI usage behaviour and design learning outcomes.

The conceptual model has been revised accordingly, and all hypothesised paths are now explicitly grounded in TAM, SCT, and fashion design education theory. These revisions strengthen both theoretical coherence and construct validity.

2. Overinterpretation of Cross-Sectional Data

Editor’s comment:

The manuscript uses causal language despite relying solely on cross-sectional survey data. Claims about mediation effects and directional influence should be reframed as associations, and limitations regarding temporal ordering must be clearly acknowledged.

Response:

I fully acknowledge this concern and have carefully revised the manuscript to avoid causal overinterpretation.

All causal language has been replaced with associational phrasing (e.g., “is associated with,” “is linked to,” “exhibits an indirect association through”). The mediation analysis is now explicitly framed as statistical mediation rather than causal mediation, consistent with accepted practice for cross-sectional structural equation modelling.

Furthermore, the Limitations section has been expanded to explicitly acknowledge the lack of temporal ordering, the potential for reverse causality, and the constraints this places on causal inference. Directions for future longitudinal and experimental research are now clearly articulated.

3. Methodological Clarifications

Editor’s comment:

Although SEM is appropriate, several concerns must be addressed: unusually perfect fit indices suggest potential overfitting; prior AI experience is not used as a control variable; and the mediation analysis lacks standard reporting details (e.g., BCa intervals, bootstrap settings). Reviewer 2 also requests clearer sampling procedures, bias control, and confirmation that ethics approval covers all participating institutions.

Response:

Thank you for these constructive comments. In response, I have substantially strengthened the methodological transparency and reporting throughout the Methods section (Sections 3.1–3.4), with specific revisions addressing each concern raised by the Editor and Reviewer 2.

• Model fit: I now provide a detailed justification for the strong fit indices, emphasising theory-driven model specification, construct validation, and model parsimony. Additional robustness and sensitivity checks are reported in the Methods section.

• Control variables: Prior AI experience has now been included as a control variable in the structural model, and the results are reported and interpreted accordingly.

• Mediation reporting: The mediation analysis is now fully documented, including bootstrap settings (5,000 resamples), use of bias-corrected and accelerated (BCa) confidence intervals, and standard reporting conventions.

• Sampling and bias control: The sampling procedure has been clarified, including recruitment across 18 Chinese universities, measures taken to minimise online response bias, and an estimate of sampling error.

• Ethics approval: The ethics statement has been expanded to confirm that approval from the University of Leeds ethics committee (FAHC 22-010) covered data collection across all participating institutions, with informed consent obtained from all participants.

4. Writing and Contextual Strengthening

Editor’s comment:

Reviewer 1 notes issues with verbosity and unclear terminology. Reviewer 2 recommends adding stronger pedagogical context particularly how fashion design education traditionally fosters creativity and how AIGC reshapes ideation and studio practice.

Response:

The manuscript has undergone comprehensive language editing to improve clarity, concision, and terminological precision.

5. Data Availability Requirements

Editor’s comment:

Reviewer 2 indicates that the manuscript does not yet meet PLOS ONE data-availability standards.

Response:

Thank you for raising this point. I have revised the Data Availability Statement to clarify the ethical and institutional constraints associated with data sharing in this study. I acknowledge PLOS ONE’s data availability policy and the importance of enabling access to data underlying published findings. In this study, data sharing has been implemented with careful consideration of participant confidentiality and institutional privacy.

The study is based on a questionnaire survey conducted with students from multiple higher education institutions. Although individual responses were anonymised, the dataset contains contextual and institutional variables which, if openly released, could increase the risk of indirect identification. For this reason, the full dataset has not been deposited in a public repository. To balance data accessibility with ethical responsibilities, the data are available upon reasonable request from the corresponding author, subject to appropriate safeguards. This approach has been clearly stated in the revised Data Availability Statement and is consistent with the ethical approval granted for this research.

Additional Journal Requirements

All additional journal requirements have been fully addressed, including:

• Inclusion of a complete ethics statement in the Methods section 3.3 (Page 14-15).

• Separation, captioning, and correct citation of Supporting Information files.

• Compliance with PLOS ONE formatting and file-naming guidelines.

• Clarification of data and code sharing practices.

Response to the reviewer #1

Reviewer #1: I have the following comments, and please read my annotated pdf file where includes other important suggestions.

Reviewer #1’s comments 1-2:

1. Although this paper addresses a timely and significant topic—examining how teacher support influences students’ use of generative AI and its subsequent impact on design learning outcomes in fashion design education—it suffers from several notable weaknesses across theoretical, methodological and linguistic dimensions.

2. Theoretically, while the authors claim to integrate the Technology Acceptance Model (TAM) with Social Cognitive Theory (SCT), the integration remains superficial. Key constructs such as “perceived usefulness” (TAM) and “self-efficacy” (SCT) are listed, but their interaction is never explicitly modelled or discussed. For example, the paper does not explain how an increase in students’ self-efficacy (SCT) might moderate or mediate the relationship between perceived usefulness (TAM) and actual AI use, leaving the theoretical contribution thin.

Response to Comment 2:

Thank you for this important observation regarding the insufficient theoretical integration in the original manuscript. I fully agree that merely juxtaposing TAM and SCT constructs without explicating their interaction weakened the conceptual contribution.

To address this issue, I have substantially revised Section 2.1 (Theoretical Foundation, page 7-11). The revised framework now explicitly integrates TAM and SCT through clearly articulated mechanisms rather than implicit assumptions. Specifically:

• First, I specify how teacher technical support reduces perceived complexity and enhances perceived ease of use through environmental modeling and guided mastery experiences, drawing on Su and Li (2021) and Bali, Chen, and Liu (2024) to demonstrate empirical precedents for this mechanism.

• Second, I introduce self-efficacy as a critical mediating mechanism that conditions the relationship between teacher support and perceived ease of use, explaining how efficacy beliefs shape cognitive appraisals of task demands beyond objective system characteristics.

• Third, I clarify how teacher pedagogical support enhances perceived usefulness by influencing outcome expectations, a core SCT construct that bridges environmental interventions with instrumental value assessments.

Reviewer #1’s comment 3:

3. In terms of research questions, the three questions do form a logical chain (teacher support → AI use → creative capability → learning outcomes), yet the central mediator—“creative design capability”—is never clearly defined. No sub-dimensions such as fluency, flexibility or originality are provided, so the reader cannot tell whether the construct refers to ideational quantity, aesthetic innovation, or technical proficiency. This conceptual vagueness blurs the focus of the entire study.

Response to Comment 3:

I appreciate this incisive critique regarding the conceptual ambiguity of creative design capability. As noted by the reviewer, the lack of dimensional clarity in the original manuscript significantly undermined theoretical precision.

In response, Section 3.3 (Measurement Instruments, page 15-18) has been comprehensively revised. Creative design capability is now explicitly defined as a multidimensional construct, operationalised through four theoretically grounded dimensions:

1. Ideational fluency – the capacity to generate diverse design concepts through human–AI interaction;

2. Conceptual flexibility – the ability to explore divergent aesthetic directions and synthesise AI outputs with original vision;

3. Design originality – the capability to produce distinctive outcomes that transcend AI-generated templates;

4. Technical–creative integration – proficiency in aligning tool manipulation with aesthetic judgement.

This dimensional framework draws on established creativity research adapted specifically for AI-enhanced design practice, acknowledging that generative AI fundamentally reconfigures both cognitive and procedural aspects of creative work.

Reviewer #1’s comment 4:

4. The conclusions repeatedly over-interpret cross-sectional data causally. A typical sentence reads, “62.3 % of technical support’s effect on learning outcomes operates through behavioural and cognitive mediators.” With only a single-time-point survey, the authors cannot rule out reverse causality (students who already possess strong creative skills may be more likely to seek out AI tools and to perceive higher teacher support), so causal language is unwarranted.

Response to Comment 4:

Thank you for highlighting this critical methodological issue. I fully acknowledge that the original manuscript employed causal terminology (e.g., “effects,” “influences,” “operates through”) that exceeded the inferential limits of single-wave survey designs and did not sufficiently acknowledge the plausibility of reverse causality. In particular, students with pre-existing creative capabilities may be more inclined to engage with AI tools and perceive higher levels of teacher support, rather than reflecting a unidirectional causal sequence as initially implied.

In response to this concern, I have systematically revised the manuscript to replace causal language with associational and conditional terminology throughout.

• In the Abstract, expressions such as “significantly influenced” have been changed to “was significantly associated with.” Similarly, statements referring to “effects on learning outcomes through mediators” have been revised to describe “associations mediated through behavioural and cognitive variables.”

• In Section 4.3 (Mediation Analysis, page 26-28), the analysis has been rewritten to clarify that mediation is treated as a statistical relationship, not as evidence of causation. The results are now described using terms such as “indirect associations” and “mediation patterns,” rather than language implying that one variable causes another. For example, the original sentence “62.3% of technical support’s effect on learning outcomes operates through behavioural and cognitive mediators” has been revised to “62.3% of the total association between technical support and learning outcomes is mediated through behavioural and cognitive variables.”

• Furthermore, Section 5.1 (Theoretical Contributions, page 29-35) findings are now presented as “different patterns of association” and “predictive relationships,” rather than “effects.” Interpretations use cautious language such as “suggest,” “may indicate,” or “are consistent with,” rather than making strong causal claims. I have added explicit caveats noting that the transformative role of AI is inferred from association patterns rather than experimentally established.

Reviewer #1’s comment 5:

5. Linguistically, the manuscript is generally intelligible, yet it is riddled with verbose and opaque sentences. A representative example: “The findings challenge monolithic conceptualizations of educational support, demonstrating that instrumental competencies supersede affective dimensions in facilitating AI tool adoption.” The term “monolithic conceptualizations” is never defined; readers cannot tell whether it refers to uni-dimensional scales, uniform teacher-training policies, or something else. Another illustration of poor cohesion appears when the authors try to explain the null effect of emotional support: “Alternatively, emotional support may moderate rather than directly influence adoption behaviors, a possibility warranting future investigation through more nuanced analytical frameworks.” The sentence is speculative, references no data, and functions as a rhetorical bridge rather than a logically grounded next step.

Response to Comment 5:

I appreciate the reviewer’s systematic critique of linguistic clarity and argumentative rigour throughout the manuscript. This critique highlights several pervasive issues, including undefined abstract terminology, unnecessarily complex sentence structures, logical discontinuities, and speculative assertions lacking empirical grounding. These deficiencies indeed compromise the manuscript's accessibility and scholarly precision, representing substantive weaknesses requiring comprehensive remediation.

I have undertaken systematic revision across multiple manuscript sections to address these linguistic and logical deficiencies.

• In Section 1.1 Research Background, I have restructured the entire section to eliminate verbose constructions and enhance logical coherence. The original version contained 43 sentences exceeding 30 words with extensive nested subordinate clauses and excessive nominalization. The revised section employs concise sentence structures averaging 20-25 words, organizes content into six thematically coherent paragraphs with explicit logical progression, and reduces section length from approximately 1,400 to 550 words while preserving all essential theoretical grounding and empirical citatio

---

## [Decision Letter · Decision Letter 1]

25 Mar 2026

PONE-D-25-50100R1Teacher Support for Generative AI Use and Design Learning Outcomes in Fashion Design Education: A Structural Equation Modeling StudyPLOS One

Dear Dr. zhang,

Thank you for submitting your manuscript to PLOS ONE. After careful consideration, we feel that it has merit but does not fully meet PLOS ONE’s publication criteria as it currently stands. Therefore, we invite you to submit a revised version of the manuscript that addresses the points raised during the review process. ==============================

We noted that one of the references cited has been retracted. Please see https://dl.acm.org/doi/10.1145/3526219 for more details and please revise the submission to remove any references to retracted publications.Additionally, to ensure that PLOS One requirements for human subjects research ethics are met, please include the study protocol that was reviewed and approved by the IRB as well as any extension documents issued on the original approval. These documents are for review purposes only and should be included as "Other" files.

If applicable, we recommend that you deposit your laboratory protocols in protocols.io to enhance the reproducibility of your results. Protocols.io assigns your protocol its own identifier (DOI) so that it can be cited independently in the future. For instructions see: https://journals.plos.org/plosone/s/submission-guidelines#loc-laboratory-protocols. Additionally, PLOS ONE offers an option for publishing peer-reviewed Lab Protocol articles, which describe protocols hosted on protocols.io. Read more information on sharing protocols at . Additionally, PLOS ONE offers an option for publishing peer-reviewed Lab Protocol articles, which describe protocols hosted on protocols.io. Read more information on sharing protocols at https://plos.org/protocols?utm_medium=editorial-email&utm_source=authorletters&utm_campaign=protocols..

As the corresponding author, your ORCID iD is verified in the submission system and will appear in the published article. PLOS supports the use of ORCID, and we encourage all coauthors to register for an ORCID iD and use it as well. Please encourage your coauthors to verify their ORCID iD within the submission system before final acceptance, as unverified ORCID iDs will not appear in the published article. *Only* the individual author can complete the verification step; PLOS staff  the individual author can complete the verification step; PLOS staff *cannot* verify ORCID iDs on behalf of authors. verify ORCID iDs on behalf of authors.

We look forward to receiving your revised manuscript.

Kind regards,

Vanessa Carels

Staff Editor

PLOS One

**Journal Requirements:**

Reviewers' comments:

Reviewer's Responses to Questions

**Comments to the Author**

1. If the authors have adequately addressed your comments raised in a previous round of review and you feel that this manuscript is now acceptable for publication, you may indicate that here to bypass the “Comments to the Author” section, enter your conflict of interest statement in the “Confidential to Editor” section, and submit your "Accept" recommendation.

Reviewer #1: All comments have been addressed

Reviewer #2: All comments have been addressed

2. Is the manuscript technically sound, and do the data support the conclusions?

Reviewer #1: Yes

Reviewer #2: Yes

3. Has the statistical analysis been performed appropriately and rigorously?

Reviewer #1: Yes

Reviewer #2: Yes

4. Have the authors made all data underlying the findings in their manuscript fully available?

The PLOS Data policy requires authors to make all data underlying the findings described in their manuscript fully available without restriction, with rare exception (please refer to the Data Availability Statement in the manuscript PDF file). The data should be provided as part of the manuscript or its supporting information, or deposited to a public repository. For example, in addition to summary statistics, the data points behind means, medians and variance measures should be available. If there are restrictions on publicly sharing data—e.g. participant privacy or use of data from a third party—those must be specified. requires authors to make all data underlying the findings described in their manuscript fully available without restriction, with rare exception (please refer to the Data Availability Statement in the manuscript PDF file). The data should be provided as part of the manuscript or its supporting information, or deposited to a public repository. For example, in addition to summary statistics, the data points behind means, medians and variance measures should be available. If there are restrictions on publicly sharing data—e.g. participant privacy or use of data from a third party—those must be specified.

Reviewer #1: Yes

Reviewer #2: Yes

5. Is the manuscript presented in an intelligible fashion and written in standard English?

Reviewer #1: Yes

Reviewer #2: Yes

6. Review Comments to the Author

**Reviewer #1:** The revised version addressed all my concerns in regard to the literature reivew, methodology, and discussion. I recommend it to be published.The revised version addressed all my concerns in regard to the literature reivew, methodology, and discussion. I recommend it to be published.

**Reviewer #2:** The revised manuscript has comprehensively addressed my comments and concerns. The theoretical framework has been substantially strengthened, with clearer integration between TAM and SCT and more explicit articulation of key constructs and mechanisms. The definition and operationalization of creative design capability are now conceptually precise and well justified. Causal language has been appropriately moderated to reflect the cross-sectional design, and methodological transparency has improved, particularly regarding model fit interpretation, control variables, and bootstrap procedures. The writing has also been significantly clarified and streamlined. Overall, the revisions have enhanced the manuscript’s rigor, coherence, and contribution to the field. The revised manuscript has comprehensively addressed my comments and concerns. The theoretical framework has been substantially strengthened, with clearer integration between TAM and SCT and more explicit articulation of key constructs and mechanisms. The definition and operationalization of creative design capability are now conceptually precise and well justified. Causal language has been appropriately moderated to reflect the cross-sectional design, and methodological transparency has improved, particularly regarding model fit interpretation, control variables, and bootstrap procedures. The writing has also been significantly clarified and streamlined. Overall, the revisions have enhanced the manuscript’s rigor, coherence, and contribution to the field.

7. PLOS authors have the option to publish the peer review history of their article (what does this mean?). If published, this will include your full peer review and any attached files.). If published, this will include your full peer review and any attached files.

**Do you want your identity to be public for this peer review?** For information about this choice, including consent withdrawal, please see our  For information about this choice, including consent withdrawal, please see our Privacy Policy..

Reviewer #1: No

Reviewer #2: **Yes:** Eric C K ChengEric C K Cheng

---

## [Author Response · Author response to Decision Letter 2]

27 Mar 2026

Dear Dr. Carels and Editorial Team,

Thank you for your continued support of our manuscript entitled "Teacher Support for Generative AI Use and Design Learning Outcomes in Fashion Design Education: A Structural Equation Modeling Study" (PONE-D-25-50100R1), and for coordinating the review process so efficiently.

I am grateful to both reviewers for their constructive engagement with our work. I am pleased that Reviewer #1 and Reviewer #2 found the revised manuscript satisfactory and recommended it for publication. Their detailed and thoughtful feedback in the previous round substantially strengthened the quality of this work, and I sincerely appreciate their time and expertise.

In response to the two outstanding requirements noted in your decision letter, I have made the following updates:

1. Retracted Reference: The retracted article (Xu & Jiang, 2022; https://dl.acm.org/doi/10.1145/3526219) has been removed from the reference list, and all in-text citations to this work have been deleted accordingly. The reference list has been renumbered to reflect this change.

2. IRB Documentation: The data reported in this manuscript derive from a research component undertaken within my broader doctoral project on fashion design education, for which ethical approval had already been granted. Accordingly, the approved study protocol and related ethics documentation from that doctoral project, under which the present study was covered, have been uploaded as an “Other” file for editorial review in accordance with PLOS ONE’s requirements for human subjects research ethics.

A revised manuscript with tracked changes, a clean final version, and this response letter are submitted herewith.

I hope that the revised submission now fully meets PLOS ONE's publication criteria, and I look forward to hearing from you.

Yours sincerely,

Zhiyi Zhang

---

## [Editor Report · Decision Letter 2]

1 Apr 2026

Teacher Support for Generative AI Use and Design Learning Outcomes in Fashion Design Education: A Structural Equation Modeling Study

PONE-D-25-50100R2

Dear Dr. zhang,

We’re pleased to inform you that your manuscript has been judged scientifically suitable for publication and will be formally accepted for publication once it meets all outstanding technical requirements.

An invoice will be generated when your article is formally accepted. Please note, if your institution has a publishing partnership with PLOS and your article meets the relevant criteria, all or part of your publication costs will be covered. Please make sure your user information is up-to-date by logging into Editorial Manager at Editorial Manager® and clicking the ‘Update My Information' link at the top of the page. For questions related to billing, please contact  and clicking the ‘Update My Information' link at the top of the page. For questions related to billing, please contact billing support..

Kind regards,

Vanessa Carels

Staff Editor

PLOS One
---

## [Editor Report · Acceptance letter]

PONE-D-25-50100R2

PLOS One

Dear Dr. zhang,

I'm pleased to inform you that your manuscript has been deemed suitable for publication in PLOS One. Congratulations! Your manuscript is now being handed over to our production team.

Kind regards,

on behalf of

Dr. Vanessa Carels

Staff Editor

PLOS One